# High-Density Lipoproteins in Kidney Disease

**DOI:** 10.3390/ijms22158201

**Published:** 2021-07-30

**Authors:** Valentina Kon, Hai-Chun Yang, Loren E. Smith, Kasey C. Vickers, MacRae F. Linton

**Affiliations:** 1Department of Pediatrics, Vanderbilt University Medical Center, Nashville, TN 37232, USA; valentina.kon@vumc.org (V.K.); haichun.yang@vumc.org (H.-C.Y.); 2Department of Pathology, Microbiology and Immunology, Vanderbilt University Medical Center, Nashville, TN 37232, USA; 3Department of Anesthesiology, Vanderbilt University Medical Center, Nashville, TN 37232, USA; loren.e.smith@vumc.org; 4Atherosclerosis Research Unit, Department of Medicine, Division of Cardiovascular Medicine, Vanderbilt University Medical Center, Nashville, TN 37232, USA; kasey.vickers@vumc.org; 5Department of Pharmacology, Vanderbilt University, Nashville, TN 37232, USA

**Keywords:** high-density lipoprotein (HDL), apolipoprotein AI, chronic kidney disease, acute kidney disease, sRNA, malondialdehyde, isolevuglandins

## Abstract

Decades of epidemiological studies have established the strong inverse relationship between high-density lipoprotein (HDL)-cholesterol concentration and cardiovascular disease. Recent evidence suggests that HDL particle functions, including anti-inflammatory and antioxidant functions, and cholesterol efflux capacity may be more strongly associated with cardiovascular disease protection than HDL cholesterol concentration. These HDL functions are also relevant in non-cardiovascular diseases, including acute and chronic kidney disease. This review examines our current understanding of the kidneys’ role in HDL metabolism and homeostasis, and the effect of kidney disease on HDL composition and functionality. Additionally, the roles of HDL particles, proteins, and small RNA cargo on kidney cell function and on the development and progression of both acute and chronic kidney disease are examined. The effect of HDL protein modification by reactive dicarbonyls, including malondialdehyde and isolevuglandin, which form adducts with apolipoprotein A-I and impair proper HDL function in kidney disease, is also explored. Finally, the potential to develop targeted therapies that increase HDL concentration or functionality to improve acute or chronic kidney disease outcomes is discussed.

## 1. A Shift in Focus from HDL Cholesterol Levels to HDL Function

The demonstration of the inverse relationship between high-density lipoprotein (HDL)-cholesterol (HDL-C) concentrations and cardiovascular events by epidemiological studies [1] stimulated a tremendous amount of research into the atheroprotective mechanisms of HDL [2]. The failure of clinical trials of HDL-C raising drugs, including niacin [3,4] and CETP inhibitors [5,6], has raised doubts about the potential of therapeutics designed to raise HDL-C as an approach to preventing cardiovascular events. In addition, Mendelian randomization studies have failed to show significant associations between genetic variants that impact HDL-C levels and the risk of cardiovascular events [7]. Furthermore, several large observational studies have shown that extremely high levels of HDL-C are associated with increased all-cause mortality, with the inflection point around 77 mg/dL in men and 97 mg/dL in women [8]. The major cause of this increased mortality, cardiovascular or non-cardiovascular, varied with the population being studied. Possible explanations for this increased risk of mortality with extreme HDL-C concentrations include genetic variants, impaired HDL (particle) function and residual confounding factors, e.g., alcohol intake [8]. Mounting evidence suggests that HDL function may be a more worthwhile therapeutic target than raising HDL-C levels. Several studies have shown that the cholesterol efflux capacity of HDL (a measure of the first step in reverse cholesterol transport) is an independent predictor of atherosclerosis and atherosclerotic cardiovascular events [9,10,11]. More recent studies have demonstrated that the HDL inflammatory index is an inverse predictor of cardiovascular events that is independent of both HDL-C levels and HDL efflux capacity [12,13]. These observations have provided strong support for the concept that metrics other than HDL-C levels, such as HDL functionality and composition, may better capture HDL cardiovascular protection.

HDL can be viewed as a pool of heterogeneous particles that vary in size and carry a diverse cargo, including a wide variety of proteins, lipids, bioactive lipids and small RNAs (sRNA), which all may influence its function [2]. The majority of studies on HDL composition and function have focused on the anti-atherogenic properties of HDL. However, many of these same HDL functions also pertain to non-cardiovascular diseases, such as anti-inflammatory and antioxidant properties of HDL. HDL composition and functions have been proposed to play important roles in non-cardiovascular diseases, including infections, inflammatory disorders and kidney disease. Chronic kidney disease is associated with dramatic changes in HDL metabolism with low levels of HDL-C and impaired HDL function [14]. Importantly, end stage renal disease (ESRD) is associated with a dramatically increased risk of death due to cardiovascular events, but, unlike other high-risk groups, treatment of subjects with ESRD on dialysis with statins does not mitigate this risk [15,16]. Hence, there is considerable interest in whether the alterations in HDL in chronic kidney disease (CKD) contribute to the progression of kidney disease and or the increased risk of cardiovascular events. Highly reactive dicarbonyl molecules generated by reactive oxygen species, including malondialdehyde (MDA) and isolevuglandins (IsoLG), rapidly adduct to HDL proteins and phospholipids, altering HDL function. We have recently shown that a small molecule dicarbonyl scavenger, 2-hydoxybenzylamine (2-HOBA) reduces dicarbonyl modification of HDL, improving its function and reducing atherosclerosis in mouse models of atherosclerosis [17]. Importantly, we have found that reactive dicarbonyl modification impacts HDL function in CKD. Furthermore, sRNA cargo on HDL is altered in CKD and may provide novel targets of therapy for CKD. Thus, HDL has multiple functions that have been proposed to play important roles in non-cardiovascular diseases, including infections, inflammatory disorders and kidney disease. We will review recent progress in understanding the roles of HDL in kidney disease, how kidney disease may impact HDL metabolism and function, and potential HDL-based therapeutic approaches for the treatment of kidney disease.

## 2. Kidney Regulation of HDL Metabolism

Kidney disease leads to abnormalities in lipids and lipoproteins. Numerous diseases affect renal vessels, glomeruli, tubules and the renal interstitium. These can be acute disorders such as acute kidney injury associated with cardiac surgery or allergic reaction to a medication, or chronic progessive scarring associated with hypertension, diabetes, or immunologic disorders such as lupus erythematosus. The extent and character of the dyslipidemia depends on the degree of kidney impairment, the underlying etiology, and whether proteinuria, especially nephrotic syndrome, is present [18,19]. CKD-related dyslipidemia is characterized by hypertriglyceridemia and depressed levels of circulating HDL-C. In addition, extensive evidence supports the notion that kidney injury modulates the structure and function of HDL. Critically, recent evidence indicates that kidneys themselves participate in metabolism of HDL and its components. An overview of this novel concept is summerized in Figure 1. The pathways in renal handling of HDL include (1) glomerular filtration, (2) tubular uptake that leads to (3) transcytosis (or catabolism) (4) transport via the renal lymphatic network back to the circulation and (5) urinary loss. As discussed below, diseases that affect the glomerular filtration barrier, tubular transporters that reabsorb the glomerular filtrate, and the structure and dynamics of the renal lymphatic vessels can modulate the level, structure, and functionality of HDL. The kidneys also affect extra-renal metabolism of HDL. Kidney diseases can affect many of these regulatory processes. In addition, renal diseases are associated with elevated levels of lipoprotein (a) (Lp(a)), which is an independent risk factor for cardiovascular events [14]. Interestingly, SR-B1 (scavenger receptor class B type 1), encoded by the gene SCARB1, is a lipoprotein receptor that binds HDL, mediating hepatic uptake of cholesterol esters from HDL, and has also been identified as a receptor for Lp(a) [20]. Furthermore, variants in the human SCARB1 gene are associated with high levels of both HDL cholesterol and Lp(a) [21].

### Renal Processes Regulating HDL Homeostasis

In the normal kidney, the glomerular capillary filtration barrier prevents the passage of molecules >60–100 kD. Mature spherical HDL3 and HDL2 subclasses exceed this mass; however, discoidal pre-β HDL (60–85 kD), apoA-I (28 kD), apoA-II (17 kD), apoA-IV (46 kD), and HDL-associated enzymes such as lecithin-cholesterol acyl transferase (LCAT) (67 kD) easily cross the normal glomerular filtration barrier. Common glomerulopathies that disrupt the glomerular filtration barrier permit filtration of greater quantity and greater variety of subclasses of HDL particles (Figure 1, step 1). Once filtered, these particles bind to the cubilin-amnionless-megalin complex on proximal tubule epithelial cells and are endocytosed via processes involving the neonatal Fc receptor (Figure 1, step 2) [22]. After endocytosis, cubilin and HDL dissociate in endo-lysosomal vesicles as HDL particles are trafficked to lysosomes for degradation (Figure 1, step 3). Notably, even a modest degree of glomerular impairment increases the fractional catabolic rate of ApoA-I and small HDL3 [23]. In addition to HDL catabolism by endocytosis, filtered HDL (and free apoA-I) can be transcytosed through the proximal tubule epithelial cells and returned to the circulation (Figure 1, steps 1–3). Cubilin deficiency and proximal tubular reabsorption failure due to Fanconi syndrome increases urinary excretion of apoA-I [22]. Cubilin-deficient mice have reduced proximal tubule uptake and increased urinary loss of apoA-I and show a significant decrease in plasma levels of apoA-I and small HDL3 [24]. The larger, more mature HDL2 particles are not detected in the urine, and the plasma HDL2 is not reduced in the cubilin-deficient mice compared with mice with intact cubilin. These results substantiate key roles for kidneys in the filtration and salvage of apoA-I/HDL particles, and thus, maintenance of HDL homeostasis. These observations also underscore that conditions that impede lipidation and maturation of HDL, or encourage protein dissociation from the mature HDL particle, likely increase HDL filtration due to reduced HDL size. Humans with genetic familial LCAT deficiency have low levels of HDL-C and develop renal disease [25]. For example, LCAT deficiency impairs maturation of preβ-HDL, and the low plasma HDL levels in affected individuals reflect increased appearance of HDL in the glomerular ultrafiltrate and tubular catabolism of immature, small HDL particles [26]. Reduced LCAT concentration and activity have been documented in nephrotic rats and have been linked to urinary losses of the enzyme [27]. Acquired LCAT deficiency has been suggested as a possible mechanism for reduced plasma HDL in moderate CKD [28], although to what extent the urinary losses contribute to reduced plasma levels is uncertain.

In addition to the cubilin-megalin complex, tubular epithelial cells express lipid and cholesterol transporters, including ATP-binding cassette transporter-A1 (ABCA1), ATP-binding cassette transporter- G1 (ABCG1), and scavenger receptor-BI (SR-B1) [29]. Currently, it is unclear how these transporters affect catabolic versus transcytosis pathways for HDL in tubular epithelial cells. However, patients with Tangier disease who express defective or deficient quantities of ABCA1 have reduced HDL biosynthesis and a dramatic reduction in the plasma residence time of HDL (0.22 days in patients with Tangier disease vs. 6 days in normal subjects). Low HDL particle concentrations have been ascribed to an increase in renal catabolism of HDL (Figure 1, step 3) [30]. In unpublished studies, we found proteinuric kidney injury significantly upregulates renal expression of the ABCA1 and SR-B1 genes and proteins. Interestingly, SR-B1 has been found to be critically important for HDL transcytosis across endothelial cells in the brain; therefore, it is possible that SR-B1 may direct HDL catabolism/transcytosis across renal tubular epithelial cells [31]. This may be particularly relevant in disease states characterized by modifications in the structure and composition of HDL that can influence the pathway of tubular reabsorption. We found that modification of apoA-I by reactive dicarbonyls, e.g., IsoLG, activated in conditions with oxidative stress such as kidney disease, increases uptake by tubular epithelial cells compared with normal apoA-I. These results raise the possibility that potentially harmful lipoproteins/HDL may be more avidly taken up and deposited in the renal interstitium. The route for apoA-I/HDL beyond the renal tubule may also be affected by apoA-I/HDL modification. Recently, we found that, like lipoprotein transport out of the interstitium of other tissues, salvaged apoA-I is detected in renal lymphatic vessels and that the renal lymph of animals with kidney injury had elevated levels of apoA-I, including IsoLG-modified apoA-I, compared with uninjured animals (Figure 1, step 4). These results are important because apoA-I/HDL are powerful stimuli for lymphangiogenesis accompanying proteinuric injury [32,33].

In addition to modulating renal parenchymal cells through direct interaction of HDL with renal parenchymal cells described above, another consequence of renal handling of HDL is that urinary appearance of HDL components can serve as markers of kidney damage (Figure 1, step 5). This is an important point because aside from albuminuria, there is current a paucity of reliable markers that can be used to diagnose renal disease or the follow the renal response to therapeutic interventions. Early stages of renal injury are associated with elevated urinary apoA-I and LCAT, likely reflecting disruption of the glomerular barrier and/or proximal tubules. We found that children with a variety of kidney abnormalities have elevated urinary excretion of apoA-I associated with underlying disease [34]. Subjects with proximal tubulopathies (Fanconi syndrome, Dent disease, Lowe syndrome, and cystinosis), renal dysplasia, glomerulonephritis, and relapsed nephrotic syndrome (NS) had the greatest elevations while those with NS in remission, nephrolithiasis, polycystic kidney disease, or hypertension were not different from controls. Aside from increased quantity of urinary apoA-I, we found that urine contained irregular high molecular weight (HMW) forms of apoA-I. All children with relapsed focal segmental glomerulosclerosis (FSGS) excreted multiple HMW forms of apoA-I that did not parallel the quantity of urinary apoA-I nor the quantity of albuminuria. These data are interesting because cross-linking of apoA-I produces HMW dimers and trimers with impaired functionality [35]. In a separate study, proteomic analysis of plasma and urine in patients with recurrent FSGS post transplantation reported apoA-I with a slightly higher molecular weight than authentic apoA-I, which differentiated recurrent FSGS from nonrecurrence [36]. Nearly all patients with recurrence had the modified apoA-I in their urine, while none had it in the serum suggesting that the intrarenal milieu in FSGS promotes apoA-I modification, and this modified apoA-I then appears in the urine. Whether it has a pathogenic role in renal injury is unknown.

## 3. Renal Impact on Extra-Renal Metabolism of HDL

In addition to the intrarenal processes of filtration/reabsorption/catabolism/excretion, kidneys modulate extrarenal production/metabolism of HDL components in the liver, plasma, and intestines. This becomes conspicuous with declining kidney function, especially when accompanied by heavy proteinuria. The mechanisms underlying the extrarenal modulation of HDL include (1) reducing hepatic lipase and decreasing hepatic extraction of HDL-triglycerides and phospholipids, (2) reducing adapter molecule PDZ-containing kidney protein 1 (PDZK1), which destabilizes hepatocyte SR-B1 and hepatic uptake of HDL-cholesterol, (3) increasing cholesterol ester transfer protein (CETP), and (4) increasing acyl-CoA cholesterol acyltransferase-1 (ACAT-1), which limits maturation of HDL [18]. Together, these kidney dysfunction-initiated changes in enzymes, receptors, and transporters and disrupted remodeling in the plasma impair normal HDL maturation, creating particles that are more susceptible to urinary loss and/or enhanced catabolism by the liver and kidneys.

## 4. Kidney Disease Modulation of HDL

### 4.1. Kidney Disease Effect on Circulating Levels and Composition of HDL

Recognition that HDL and its components are filtered then catabolized, salvaged or excreted indicates that diseases of the kidneys that impair these processes impact the concentration, composition, and functionality of HDL particles. A summary of this concept is illustrated in Figure 2. The most common changes documented in HDL particles isolated from CKD patients (HDL^CKD^) include reduced levels of apoA-I, apoA-II, apoM, PON-1, and higher levels of serum amyloid A (SAA), apoC-II, apoC-III, apoA-IV, albumin, lipoprotein-associated phospholipase A2 (Lp-PLA2), surfactant protein B (SP-B), and α1-microglobulin/bikunin precursor [19,37,38,39,40,41,42,43]. Progressive deterioration in kidney function is paralleled by progressive disruption in the HDL proteome. Every 15 mL/min per 1.73 m^2^ decline in GFR leads to a significant increase in retinol binding protein 4 and apoC-III, and a decrease in apoL1, CETP and vitronectin [41]. Further deterioration in kidney function requiring initiation of dialysis also changes the HDL proteome. Thus, patients starting dialysis have elevated markers of inflammatory, atherosclerotic, and lipid metabolism pathways, including SAA1, SAA2, hemoglobin-b, haptoglobin-related protein, CETP, PLTP and apoE. Among the changes in the HDL^CKD^ proteome, increased SAA is among the most consistent. Indeed, SAA levels have been shown to be inversely correlated with HDL’s anti-inflammatory potency and linked to activation of formyl-peptide receptor 2 [40]. The mechanism for this detrimental effect is that SAA displaces both apoA-I and PON1 reducing the anti-oxidative and anti-inflammatory activity of SAA-enriched HDL [39]. The harmful effects are especially prominent when SAA constitutes >50% of the total protein in HDL. Symmetric dimethylarginine (SDMA) and its structural isomer of asymmetric dimethylarginine (ADMA) are endogenous products of protein methylation linked to cardiovascular disease risk that have also been shown to accumulate as kidney function decreases. SDMA-containing HDL from CKD patients interacts with endothelial Toll-like receptor 2 to enhanced NADP-dependent production of reactive oxygen species (ROS) while inhibiting endothelial nitric oxide (NO) bioavailability [44]. In mice, SDMA in HDL causes hypertension and impaired reendothelialization after carotid injury. HDL-associated SDMA from children with CKD inhibits NO synthesis, promotes superoxide production, increases expression of vascular cell adhesion molecule 1 in aortic endothelial cells, and suppresses macrophage cholesterol efflux [45]. CKD also alters the lipid composition of HDL, increasing triglycerides and lysophospholipids while reducing phospholipids and cholesterol.

In addition to changes in HDL composition, CKD causes post-translational modifications of HDL [19]. CKD increases myeloperoxidase (MPO) levels and activity. MPO is a heme protein released by leukocytes that generates ROS. MPO binds and significantly alters HDL functionality including reducing ABCA1-mediated cholesterol efflux, activating LCAT, and reducing endothelial cell survival [46]. MPO-catalyzed lipoprotein carbamylation involves formation of cyanate (a product of urea) and ε-carbamyl-lysine homocitrulline (HCit). Serum HCit and carbamylated albumin predict mortality in dialysis patients [47]. Oxidative stress at all stages of CKD also increases reactive lipid aldehydes such as α—ketoaldehydes, including the very reactive IsoLG [48]. IsoLG adducts lysine residues of proteins, which crosslink and alter protein function. We have shown that HDL particles become dysfunctional by apoA-I modification with IsoLG, which impairs apoA-I/HDL capacity to facilitate cholesterol efflux from macrophages and not only reduces HDL’s ability to inhibit cytokine induction but also potentiates LPS-induced IL-1b expression [35]. Our unpublished data show that proteinuric patients and animal models have increased urinary levels of IsoLG that is associated with urinary apoA-I and not urinary albumin. Moreover, as noted above, IsoLG-modified apoA-I is more avidly taken up by proximal tubules than unmodified apoA-I, possibly amplifying harmful effects of IsoLG-apoA-I in the renal interstitium, including profibrotic remodeling. Posttranslational modification by glycation is another mechanism producing dysfunctional HDL^CKD^, especially when CKD is associated with diabetes, which further reduces LCAT cholesterol esterification and PON1 levels [49]. The current consensus is that although CKD alters the proteome and lipidome of HDL, no known footprint for the particular constituents of HDL^CKD^ incontrovertibly predicts hard clinical endpoints such as CVD, mortality, or CKD progression. On the other hand, there is ample evidence that HDL^CKD^ functionality is altered.

### 4.2. Kidney Disease Effect on HDL Functionality

HDL cellular cholesterol efflux capacity (CEC), a central process in reverse cholesterol transport, is impaired across the spectrum of CKD [50,51]. Low CEC in CKD is associated with decreased apoA-I, apoA-II, and phospholipids, and increased apoC-III and SAA (all factors known to modulate CEC) [52]. Critically, reduction in CEC is not necessarily synchronized with dysfunction of other HDL functionalities. Indeed, as noted in the introduction, HDL inflammatory index appears to be an inverse predictor of cardiovascular events that is independent of efflux capacity and HDL-C [12,13]. We and others have shown that children with CKD or ESRD requiring dialysis who do not have long-standing comorbidities or risk factors characteristic of adults with CKD (e.g., diabetes, obesity, pre-existing cardiovascular disease) have HDL with reduced functionality including profound impairment in anti-inflammatory, anti-oxidative, and endothelial protection functions without CEC impairment [44,45,53]. Further support for the importance of non-CEC functions of HDL comes from studies showing that while renal transplantation and recovery of renal function (eGFR ~50 mL/min) improves endothelial and vascular reactivity, CEC remains depressed [54]. Even after stratification of the transplant recipients into those with good versus poor graft function, CEC remained profoundly depressed in both transplant groups and was not different than in ESRD patients requiring hemodialysis. Interestingly, although cholesterol efflux capacity has most commonly been applied to assess the risk of atherosclerotic cardiovascular disease, a prospective study examined whether cholesterol efflux capacity could predict adverse cardiovascular and renal events in recipients of renal transplants [55] found baseline CEC did not predict cardiovascular mortality or all-cause mortality. There was a strong inverse association between graft failure and efflux capacity, however, independent of circulating levels of apoA-I and HDL-C, reinforcing the idea that HDL functionality is superior to HDL-C levels in predicting adverse events, even those beyond atherosclerosis, such as renal graft loss. Whether other HDL functionalities, e.g., HDL inflammatory index predict graft loss is currently unknown.

### 4.3. Kidney Disease and HDL-sRNAs

In addition to dozens of proteins and small molecules, HDL also transports a diverse array of nucleic acids. We have previously reported that HDL transports microRNAs (miRNA) and many other classes of small non-coding RNAs [56,57]. The sRNA cargo on HDL is generally single-stranded, <50 nts in length, and fragments of longer parent transcripts. HDL transports sRNAs-derived from transfer RNAs (tRNA), ribosomal RNAs (rRNA), small nuclear RNAs (snRNA), long non-coding RNAs (lncRNA), and other transcripts [57]. Similar to other carriers of extracellular RNA, circulating HDL-sRNAs have been explored as potential disease biomarkers. The most well-studied class of sRNAs on HDL are miRNAs. Within cells, miRNAs control many genes involved in HDL biogenesis, cholesterol efflux, and uptake by the liver, including ABCA1, ABCG1, and SR-B1 [58]. Extracellular miRNAs have been explored as potential disease biomarkers. The HDL-miRNA profile has been reported to be significantly altered in multiple conditions of atherosclerotic cardiovascular diseases [59], including familial hypercholesterolemia [56], acute coronary syndrome [60], and vulnerable coronary artery disease (CAD) [61]. Metabolic status is also likely a driver of HDL-miRNA changes, as levels of miR-92a-3p, miR-223-3p, miR-122-5p on HDL were significantly increased in subjects with acute coronary syndrome and hyperglycemia [60]. HDL-miRNA changes have also been reported in kidney diseases [62,63]. Diabetic nephropathy is the major cause of progressive kidney damage and ESRD requiring dialysis and transplantation [64]. HDL-miR-132 levels were found to be significantly decreased in diabetic subjects with diabetic nephropathy (DN, eGFR > 30 mL/min/1.73 m^2^) compared to healthy controls [65]. Interestingly, miR-132 levels in whole plasma were found to be increased in DN subjects, suggesting that plasma miRNAs levels are not indicative of HDL-miRNA levels and that miRNAs on HDL are more closely linked to kidney disease than whole plasma. HDL-miR-132 levels showed a significant inverse correlation with angiopoietin 2 (ANG2) levels, a marker of microvascular injury, whereas plasma miR-132 levels were not correlated to ANG2 levels [65]. There are many reports of increased and decreased miRNAs in clinical and experimental models of kidney disease, however, direct impact of glomerular dysfunction/CKD on circulating HDL-sRNA levels is uncertain and requires further sequencing analyses [62,63].

HDL-sRNAs also have biological functions beyond disease biomarkers, as we and others have demonstrated. HDL have the capacity to delivery functional miRNAs to recipient cells where they regulate target gene expression and cellular activities [56]. For example, we have reported that HDL’s anti-inflammatory capacity is mediated, in part, through its ability to deliver miR-223-3p to recipient human coronary artery endothelial cells [66]. Currently, little is understood about whether HDL delivers miRNAs or other sRNAs to recipient podocytes or renal tubule epithelial cells which are of critical importance in renal response to injury. Based on in vitro studies, HDL was found to deliver miR-132 to human umbilical vasculature endothelial cells (HUVEC) by multiple methods [65]. HDL transferred miR-132 was observed to silence p120RasGap, a known miR-132 target gene [67]. HDL-miR-132 was found to control endothelial cell tube formation, as cells treated with HDL-miR-132 were found to have increased angiogenic activity, i.e., total tube length [65]. Therefore, loss of miR-132 levels on HDL in diabetes and/or DN may result in reduced capacity of HDL to stimulate angiogenesis in endothelial cells. Such diminution in the endothelial cell response may then compromise recovery following kidney injury. Therefore, decreased HDL function related to miRNA transport and delivery may contribute to DN and other kidney diseases.

## 5. HDL Modulates Renal Parenchymal Cells and Function

HDL effects have been most extensively studied in cells directly involved in atherosclerosis (e.g., macrophages and vascular endothelial cells). However, accumulating evidence indicates HDL affects numerous cells, including cells in the renal parenchyma (e.g., glomerular podocytes and mesangial cells, proximal tubular epithelial, and renal lymphatic endothelial cells). Since kidney disease is known to disrupt HDL composition and functionality it is possible that the impairment in cholesterol efflux, anti-inflammation, anti-oxidation, and anti-apoptosis capacities of HDL documented in macrophages may also affect renal cells and impact the renal response to injury [44,45,53]. Studies support this direct effect, showing that viability, migration, and ROS production in podocytes injured in vitro can be lessened by supplementation of normal apoA-I, HDL, or an apoA-I mimetic, 4F [68]. In vivo, 4F treatment significantly lessened proteinuria and preserved podocyte expression of glomerular synaptopodin and cell density in the NEP25 podocyte injury model of proteinuric kidney disease [68]. In the mouse IgA nephropathy model, glomerular mesangial cell proliferation and matrix expansion was suppressed by apoM, an effect that was mediated by ApoM-bounded S1P [69]. ApoM/HDL or albumin chaperoned sphingosine-1-phosphate (S1P) activated the S1P receptor 1 (S1PR1) abundant in endothelial cells and caused a reduction in vascular inflammation [70]. Injection of recombinant ApoM-Fc reduced age-related renal fibrosis by stimulating endothelial S1PR1 signaling [71]. Plasma apoM levels are depressed in CKD patients, but HDL-S1P content is increased [72,73]. Cultured mesangial cells as well as tubular epithelial cells (TEC) exposed to oxidized HDL showed increased ROS and stimulated production of pro-inflammatory factors via CD36 with activated MAPK and NF-κB pathways [74,75]. These findings complement our finding that IsoLG-modified apoA-I is more avidly taken up by TEC than normal apoA-I in association with upregulated expression of ABCA1, SR-B1, inflammation markers, and the lysosomal marker LAMP-1.

In addition to regulating glomerular and tubular cells, apoA-I/HDL modulates the phenotype of lymphatic endothelial cells and growth of the lymphatic vascular network [76,77]. S1P/S1PR1 signaling is involved in the regulation of lymphangiogenic and inflammation-related gene expression in lymphatic endothelial cell (LECs) [78]. We found that IsoLG-modified apoA-I stimulated higher expression of inflammation markers (IL-6, IL-23, CCL21) and S1P related genes (SPHK2, SPNS2) in cultured LECs than unmodified apoA-I. Moreover, ex vivo studies of the renal collecting vessels revealed IsoLG-modified apoA-I causes greater lymphatic constriction than native apoA-I. This is relevant because while kidney lymphatic vessels are normally inconspicuous, they become prominent in numerous renal diseases and may be instrumental in determining the renal response to injury and the long-term fibrotic consequences of the injury [79]. It is therefore possible that the reabsorbed filtered apoA-I/HDL directs this renal response to injury. Together, these data indicate that in the kidney, apoA-I/HDL particles and modified HDL components are filtered by the glomerulus and taken up into the renal parenchyma where they are uniquely positioned to have profound physiologic and pathophysiologic consequences on the phenotype and functionality of renal cells.

## 6. Targeting Abnormal Levels and Functionality of HDL to Benefit Renal Disease

Many, although not all, clinical studies report an inverse association between the incidence and progression of kidney disease and HDL-C. Low levels of HDL cholesterol have been associated with increased risk of kidney disease or impaired kidney function in several large observational cohort studies [80,81,82]. In addition, Mendelian randomization studies have supported a causal relationship between increased HDL cholesterol levels and better kidney function [83,84]. Low HDL-C is inversely associated with in-hospital mortality and the development of multiorgan dysfunction, including acute kidney injury (AKI) [85,86,87]. Further, low HDL-C at presentation is a stronger risk factor for sepsis-associated AKI than shock requiring vasopressors [87]. Additionally, low plasma concentrations of HDL due to a rare variant of cholesterol ester transfer protein (CETP), rs1800777, which increases cholesterol removal from HDL particles thereby increasing HDL catabolism significantly, is associated with an increased risk of developing sepsis-induced AKI [88]. In humans, higher preoperative HDL-C was associated with less AKI following surgery, an effect that was independent of other AKI risk factors [89]. Recently, we showed that higher concentrations of HDL particles, particularly small HDL particles, are associated with higher PON-1 activity, lower postoperative isofuran concentrations, and lower odds of AKI after cardiac surgery [90]. These observations support the possibility that intervening with exogenous small antioxidant HDL particles to acutely increase levels of beneficial HDL particles can lessen incidence and severity of AKI. The protection may reflect filtration, then reabsorption of the exogenous HDL that allows interaction between the therapeutic particles and renal parenchymal cells (Figure 1, steps 1 and 2). This strategy is currently under development and clinical testing [87]. The relationship between HDL-C and progressive chronic kidney disease is more controversial [91,92,93,94,95,96,97,98]. For instance, higher HDL cholesterol levels were associated with lower risk of eGFR decline in patients treated with atorvastatin [99]. However, as in the general population, a U-shaped association between HDL-C and all-cause and cardiovascular mortality has been documented in patients with advanced CKD requiring dialysis [100]. Progressive CKD has been linked to low and high HDL-C suggesting that, as in the general population, CKD-induced qualitative changes in HDL composition, HDL particle number (HDL-P), or HDL functions may be better predictors of risk for adverse consequences than HDL-C levels.

Although causality between HDL-C and renal function are difficult to show in clinical studies, animal studies demonstrate that targeting apoA-I/HDL can ameliorate kidney injury. Figure 2 shows possible therapeutic interventions that target renal handling of HDL. Compared with wild-type, apoA-I overexpressing mice had high HDL concentrations and lower serum creatinine following lipopolysaccharide-induced kidney damage [101]. Supplementation of normal apoA-I or HDL also lessened renal ischemia/reperfusion injury with improved glomerular filtration rate, decreased tubular injury, and less tubulointerstitial fibrosis [102]. The mechanism for these protective effects involves reduction in the expression of adhesion molecules, resulting in reduction in polymorphonuclear leukocyte infiltration and a reduction in oxidative stress within the renal parenchyma (Figure 1, steps 3 and 4). In another study, pretreatment with normal HDL decreased serum amyloid A (SAA) pro-inflammatory activity and prevented glomerular injury in a mouse model [103]. HDL can also bind F2-isoprostanes, potentially sequestering them from the plasma. Since F2-isoprostanes can directly affect renal vascular tone and stimulate eNOS activity, it is possible that HDL may ameliorate renal damage through F2-isoprostane sequestration [104,105,106]. Additionally, studies have shown that supplementation with an apoA-I mimetic reduced proteinuria, glomerular, and tubulointerstitial injury in proteinuric Nphs1-hCD25 transgenic (NEP25^+^) apoE^−/−^ mice [68]. Together, these data indicate that normal apoA-I/HDL can benefit injured kidneys. A phase II clinical trial is underway examining the effect of CSL112, an intravenous preparation of apoA-I developed to enhance cholesterol efflux, in patients suffering from acute myocardial infarction, including those with moderate renal impairment [107,108]. Whether CSL112 will also lessen kidney injury in this cohort will need further study.

In addition to supplementation with normal apoA-I/HDL, targeting lipoprotein modifications or cargo is another approach to improve apoA-I/HDL functionality and thus lessen the detrimental effects on renal parenchymal cells. For example, pentylpyridoxamine (PPM) reduces IsoLG modification of apoA-I/HDL and has been shown to significantly reduce experimental proteinuria, tubular injury, and lymphangiogenesis (Figure 1, steps 2–5) [68]. In human subjects, we showed that IL-1 blockade improved HDL functionality in patients with pre-dialysis CKD as well as individuals on maintenance hemodialysis [109]. Specifically, the therapeutic intervention improved HDL anti-inflammatory and antioxidative functions and reduced cellular expression of the NOD-like receptor protein (NLRP3) component of the inflammasome, a cytosolic multiprotein complex controlled by interleukin 1b. The therapy did not affect cholesterol efflux capacity. Another study showed that while both angiotensin converting enzyme inhibition (ACEI) and angiotensin receptor antagonisms (ARB) stabilized HDL cholesterol acceptor function and sustained cellular anti-oxidative effects, they did not improve anti-inflammatory effects [110]. Indeed, ACEI-treatment instead amplified the HDL inflammatory response. It is of particular interest that IL-1 blockade improved HDL functionality in patients with pre-dialysis CKD as well as individuals on maintenance hemodialysis [109]. These findings again underscore the idea that a generalized attenuation in HDL functionality in a disease setting may reflect distinct structural or biochemical changes in HDL particles that may or may not lead to concurrent dysfunctions in HDL.

## 7. Summary and Conclusions

Although liver, skeletal muscle, adipose tissue and macrophages are considered the primary sites of lipoprotein metabolism, accumulating evidence points to kidneys as critical participants in HDL homeostasis affecting concentrations, composition, and functionality of HDL particles. Aside from cholesterol efflux, HDL is now firmly established as being critical in numerous physiologic and pathophysiologic responses such as cell cycle, differentiation, proliferation, apoptosis, inflammation, and oxidative stress responses relevant to the structure/function of every organ. Thus, the primary target of HDL has expanded beyond the cardiovascular system and now includes the kidneys. As in other tissues, interactions between normal apoA-I/HDL and kidney cells provides beneficial effects while appearance of modified apoA-I/HDL in the glomerular filtrate enhances tubular uptake of these harmful lipoproteins and activates tubular and interstitial cells, which in turn initiates and perpetuates damaging responses in the kidneys. Indeed, modified lipoproteins have been proposed as a new class of uremic toxins. Currently, there is insufficient understanding of which structural, compositional, or functional alterations in HDL particles are most relevant, but advances in this area could not only improve outcomes in cardiovascular conditions but also in acute and progressive kidney diseases.

## Figures and Tables

**Figure 1 ijms-22-08201-f001:**
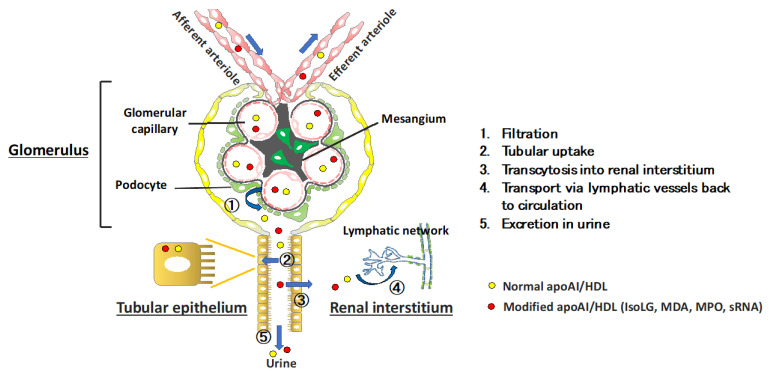
Kidney handling of normal and modified apoAI/HDL by the glomerulus, tubule epithelium, and renal interstitium involves (1) filtration (2) tubular uptake (3) transcytosis (4) transport by lymphatic vascular network in interstitium and (5) urinary excretion.

**Figure 2 ijms-22-08201-f002:**
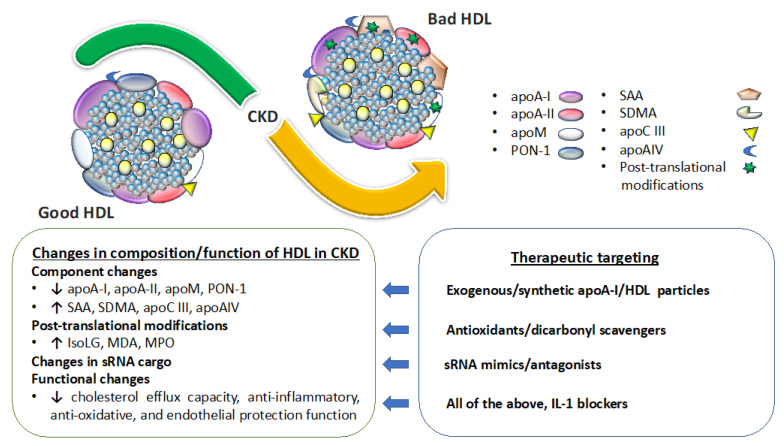
The effects of CKD on HDL structure and function, and the therapeutic implications.

## Data Availability

Data sharing not applicable. No new data were created or analyzed in this study. Data sharing is not applicable to this article.

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
