# Peer review of "High-Density Lipoproteins in Kidney Disease"

_ijms, 2021, doi:10.3390/ijms22158201_

Round 1
Reviewer 1 Report
Kon et al in their review article, “High-Density Lipoproteins in Kidney Disease” discuss the effect of HDL structure and function in HDL functions are also relevant in non-cardiovascular diseases which includes acute and chronic kidney disease. The review is thorough, well organized, comprehensively describes the relevant topics, and is pleasant to read. This review will be an important contribution to the field of HDL in Kidney diseases and I can happily recommend its acceptance for publication with some minor additions:
- The authors should add a relevant pictorial representation that summarizes the part 4. (Kidney disease modulation of HDL) This would help the readers understand this part even better.
- Moreover, adding a table with the possible targets and studies showing the relevant effects of HDL on Kidney function will be a good addition and make the review more reader friendly (information in section 5 and 6)
- I see in section 5. HDL modulates renal parenchymal cells and function (line 322). There is a subheading 5.1. HDL regulates renal cells. I fail to understand what purpose does it serve and if its needed as there are no further subheadings.
Reviewer 2 Report
Excellent rewiew.
Reviewer 3 Report
The manuscript of Kon et al. entitled “High-Density Lipoproteins in Kidney disease” reviews the role of HDL in the kidney and kidney diseases.
I would have the following suggestions to improve the manuscript:
- A short introduction of CKD (etiology, classification, etc.) would be helpful for the general audience.
- The section of lins 83-96 is a summary of the following aspects rather than an introduction towards the subject.
- Figure 1 needs to be improved considerably. For example, It should be made more clear what is visualized (e.g. mention Glomerulus; legend of cell types). The 5 different steps should be clearly visualized. The presence of IsoLG-modified HDL is unclear. Legend is rather vague.
- The various different subtypes mentioned in line 100-101 will be confusing for a general audience. An introduction of these subtypes (e.g. Table/Figure) is needed in the introduction paragraph.
- Mentioning the 5 different steps in section 2.1 (e.g. add numbers) would improve the readability. The different topics/statements in this section should be linked better as it now reads rather like a bullet-point summary.
- Section of Line 156-176 is rather pathological, while rest of this chapter is physiological and should therefore be relocated of placed into separate section.
- Chapter 3. A figure would benefit this section.
- Chapter 4.1. Readability should be improved (rather bullet-point character at the moment). A small figure or table would also help here.
- Chapter 6. A table or figure summarizing the therapeutical approaches would be helpful.
- There a quite some grammar mistakes, which should be addressed.
Round 2
Reviewer 3 Report
The authors addressed all of my concerns appropriately and I can recommend acceptance in its current form.